# Endometrial Carcinoma: Molecular Cytogenetics and Transcriptomic Profile

**DOI:** 10.3390/cancers14143536

**Published:** 2022-07-20

**Authors:** Marta Brunetti, Ioannis Panagopoulos, Valeria Vitelli, Kristin Andersen, Tarjei S. Hveem, Ben Davidson, Ane Gerda Z. Eriksson, Pernille Kristina Bjerre Trent, Sverre Heim, Francesca Micci

**Affiliations:** 1Section for Cancer Cytogenetics, Institute for Cancer Genetics and Informatics, The Norwegian Radium Hospital, Oslo University Hospital, 0379 Oslo, Norway; brunetti.marta90@gmail.com (M.B.); ioannis.panagopoulos@rr-research.no (I.P.); kad@ous-hf.no (K.A.); sverre.heim@medisin.uio.no (S.H.); 2Oslo Center for Biostatistics and Epidemiology, Department of Biostatistics, University of Oslo, 0315 Oslo, Norway; valeria.vitelli@medisin.uio.no; 3Section for Applied Informatics, Institute for Cancer Genetics and Informatics, The Norwegian Radium Hospital, Oslo University Hospital, 0379 Oslo, Norway; tarjei@icgi.no; 4Department of Pathology, The Norwegian Radium Hospital, Oslo University Hospital, 0379 Oslo, Norway; ben.davidson@medisin.uio.no; 5Institute of Clinical Medicine, Faculty of Medicine, University of Oslo, 0315 Oslo, Norway; uxbjpa@ous-hf.no; 6Department of Gynecological Oncology, The Norwegian Radium Hospital, Oslo University Hospital, 0379 Oslo, Norway; aneeri@ous-hf.no

**Keywords:** endometrial carcinoma, karyotype, chromosome 1, mutation analysis, microsatellite instability, gene expression, cancer pathways, cancer immune profile, miRNA

## Abstract

**Simple Summary:**

Endometrial carcinomas (ECs), the most frequent type of uterine body cancer, are highly heterogeneous with overlapping clinical, pathological, and molecular features. This study aimed to gain insights into these cancers’ chromosomal and genomic aberration patterns as well as their gene and miRNA expression profiles, all of which are both pathogenetically and clinically important disease features. We found that chromosome 1 was the most frequently rearranged chromosome, mostly leading to gain of 1q, that the genes *PTEN*, *PDGFRA*, *PIK3CA*, and *KIT* were the most frequent pathogenic variants, and that some other genes and miRNAs of known importance in carcinogenesis and the immune response showed consistent deregulation. This study confirms that a high degree of genetic heterogeneity characterizes EC tumors but highlights the nonrandom involvement of some loci.

**Abstract:**

Endometrial carcinomas (ECs) are histologically classified as endometrioid and nonendometrioid tumors, with each subgroup displaying different molecular profiles and clinical outcomes. Considerable biological and clinical heterogeneity exists within this scheme, however, reflecting its imperfection. We aimed to gather additional data that might help clarify the tumors’ pathogenesis and contribute toward a more meaningful classification scheme. In total, 33 ECs were examined for the presence of chromosomal aberrations, genomic imbalances, pathogenic variants, microsatellite instability, and expression profiles at both gene and miRNA levels. Chromosome 1 was the most frequently rearranged chromosome, showing a gain of all or part of the long arm. Pathogenic variants were found for *PTEN* (53%), *PDGFRA* (37%), *PIK3CA* (34%), and *KIT* (31%). High microsatellite instability was identified in 15 ECs. Comparing tumors and controls, we identified 23 differentially expressed genes of known importance in carcinogenesis, 15 genes involved in innate and adaptative immune responses, and altered expression of 7 miRNAs. miR-32-5p was the most upregulated. Our series showed a high degree of heterogeneity. Tumors were well-separated from controls, but there was no clear-cut separation between endometrioid and nonendometrioid ECs. Whether this means that the current phenotypic classification is of little relevance or if one still has not detected which genomic parameters to enter into correlation analyses remains unknown.

## 1. Introduction

Endometrial cancer is the sixth most common cancer in women and the second most commonly diagnosed cancer of female genital organs [1]. In 2020, around 417,000 new cases were detected, and 97,000 women died worldwide from the disease [1,2]. The most frequent uterine cancers are endometrial carcinomas (ECs), which account for 90% of uterine malignancies in post-menopausal women [3]. 

ECs arise from the lining of the uterus and are usually divided into two types: endometrioid, which is estrogen-associated and makes up 80% of all endometrial cancers, and hormone-independent nonendometrioid ECs (20%) [4], which can be further subdivided as serous carcinomas, clear-cell carcinomas, and carcinosarcomas. According to the World Health Organization (WHO) [3], endometrioid ECs can be low-grade (grades 1 and 2) tumors that are generally associated with a good prognosis or high-grade carcinomas carrying an intermediate prognosis [5]. Across and within the boundaries of these classifications, ECs display considerable heterogeneity, with overlapping clinical, pathological, and molecular features [6,7,8]. 

In recent years, the increasing integration of molecular and morphological tumor data has led to a better stratification of EC patients based on a deeper understanding of the carcinogenic process [9]. The Cancer Genome Atlas (TCGA) consortium performed an integrated genomic characterization of both types of ECs, identifying four molecular subgroups among endometrioid ECs: (1) ECs with *POLE* exonuclease domain mutations; (2) ECs with microsatellite instability (MSI); (3) ECs with low or few copy number alterations (CNAs); and (4) ECs with high CNA and *TP53* mutations [10,11]. These molecular features were included in the subclassification of endometrioid ECs suggested by the most recent (2020) edition of the WHO classification of female genital tract tumors [3]. Notably, karyotypic information is not part of the TGCA approach to EC, although cytogenetic data on more than 100 such tumors exist in the relevant literature [12]. 

In this paper, we report newly generated data on the karyotypic features, genomic imbalances, pathogenic variants, microsatellite instability, and expression profiles at the gene and miRNA level for 33 ECs from which clinical and pathological information was also available.

## 2. Materials and Methods

### 2.1. Tumor Material

The material consisted of 33 ECs, of which 24 were endometrioid, and 9 were nonendometrioid. The nine nonendometrioid ECs included three serous, one clear cell, three of mixed histology (endometrioid-serous, endometrioid-clear cell, and endometrioid-unclassified), and two tumors classified as “adenocarcinoma not otherwise specified (NOS)”. All were surgically removed at the Norwegian Radium Hospital between 2013 and 2017. An overview of the clinical and pathological data is given in Table 1. Six commercially available normal controls were included in the series. Four of them were total RNAs from the normal uterus of single donors (AMS Biotechnology, Abingdon, United Kingdom). The fifth was a total RNA consisting of a pool of uterine RNA from five donors (AMS Biotechnology). The sixth was from normal endometrium from a single donor (OriGene, Rockville, MD, USA).

The study was approved by the Regional Committee for Medical and Health Research Ethics (REK, project number 2011/2071; http://helseforskning.etikkom.no; accessed on 15 May 2021).

### 2.2. G-Banding and Karyotyping

Fresh tissue from a representative area of the tumors was mechanically and enzymatically disaggregated with collagenase II (Worthington, Freehold, NJ, USA). The cells were cultured and harvested using standard techniques [13]. Chromosome preparations were G-banded with Wright’s stain (Sigma-Aldrich, St Louis, MO, USA). Metaphases were analyzed, and karyograms were prepared using the CytoVision computer-assisted karyotyping system (Leica Biosystems, Newcastle, UK). The karyotypes were written according to the International System for Human Cytogenomic Nomenclature [14].

### 2.3. DNA and RNA Extraction

Fresh-frozen material from a representative area of the tumors was used to extract DNA and RNA. DNA was extracted using the Maxwell 16 extractor (Promega, Madison, WI, USA) and purified using a Maxwell 16 Cell DNA Purification kit (Promega) according to the manufacturer’s recommendations. RNA was extracted using a miRNeasy kit (QIAGEN, Hilden, Germany). The concentrations were measured using a QIAxel microfluidic UV/VIS spectrophotometer (QIAGEN) and a Quantus fluorometer (Promega). RNA quality was assessed with an Agilent RNA 6000 Nano total kit on an Agilent 2100 Bioanalyzer (Agilent Technologies, Santa Clara, CA, USA).

### 2.4. Genomic Imbalances

A whole-genome investigation was performed by means of array comparative genomic hybridization (aCGH) using the CytoSure Consortium Cancer + Single Nucleotide Polymorphism (SNP) arrays (Oxford Gene Technology, Oxford, UK) according to the manufacturers’ recommendations. Data were analyzed using Agilent Feature Extraction Software (Agilent Technologies; version 10.7.3.1) and CytoSure Interpret Software (version 4.9.40, Oxford Gene Technology). The genomic imbalances were identified using the Circular Binary Segmentation (CBS) algorithm and adding a custom-made aberration filter defining copy number alteration (CNA) as a region with a minimum of five probes gained/lost [15]. Only genomic imbalances larger than 5 Mb were scored. Annotations are based on human reference sequence GRCh37/hg19 [16].

### 2.5. Pathogenic Variants

All tumors were investigated using next-generation sequencing (NGS) with the Ion Torrent S5 platform and an Ion Ampliseq Cancer Hotspot Panel v2 Chef-Ready kit according to the company’s protocols (ThermoFisher Scientific, Waltham, MA, USA). Amplified libraries were quantified using a Quantus fluorometer and a High-Sensitivity Quantus Assay kit (Promega).

All the detected pathogenic variants were then searched for in the COSMIC database (Catalogue of Somatic Mutations in Cancer, at http://cancer.sanger.ac.uk/cosmic; accessed on 15 January 2022; version 92). Pathogenic variant analysis was performed of the five most common hotspots of the DNA polymerase epsilon, the catalytic subunit (*POLE*) gene, namely positions c.857C > G, c.1231G > T, c.890 C > T, c.1366G > C, and c.1376C > T in exons 9, 12, and 13, respectively, using M13-linked PCR primers designed to flank and amplify targeted sequences (Table 2) and a BigDye Direct Cycle Sequencing kit (ThermoFisher Scientific) according to company’s recommendations. The thermal cycling for *POLE* (exons 9 and 13) included an initial step at 95 °C for 10 min, followed by 35 cycles at 96 °C for 3 sec, 58 °C for 15 sec, 30 sec at 68 °C, and a final step at 72 °C for 2 min. The thermal cycling for exon 12 of *POLE* included an initial step at 95 °C for 10 min, followed by 35 cycles at 96 °C for 3 sec, 62 °C for 15 sec, 30 sec at 68 °C, and a final step at 72 °C for 2 min. Sequencing was performed using an Applied Biosystems SeqStudio Genetic Analyzer system (ThermoFisher Scientific). The basic local alignment search tool (BLAST; https://blast.ncbi.nlm.nih.gov/Blast.cgi; accessed on 15 January 2022) and BLAT-like alignment tool (BLAT; https://genome-euro.ucsc.edu/cgi-bin/hgBlat; accessed on 15 January 2022) programs were used for the computer analysis of sequence data.

### 2.6. Microsatellite Instability (MSI) Status

The tumors’ MSI status was investigated using an Applied Biosystems TrueMark MSI Assay (ThermoFisher Scientific) using 13 markers according to the manufacturer’s instructions. Negative and positive controls were used to evaluate the efficiency of the amplification step. Fragment analysis was performed using an Applied Biosystems SeqStudio Genetic Analyzer. The data were analyzed using the Applied Biosystem TrueMark MSI Analysis Software (ThermoFisher Scientific).

The samples were considered MS stable (MSS), MSI low (MSI-L), or MSI high (MSI-H) if none, one, or >30% of examined markers showed instability.

### 2.7. Messenger RNA (mRNA) and MicroRNA (miRNA) Expression Profile

Gene transcripts (mRNAs) and miRNA expression analyses were performed on 33 ECs and 6 normal tissue samples with the help of the nCounter SPRINT NanoString platform (NanoString Technologies, Inc., Seattle, WA, USA) using nCounter PanCancer Pathway, PanCancer Immune Profiling, and the nCounter Human v3 miRNA Expression Assay kit. For gene expression, 25 ng of total RNA from each sample was hybridized with the code set at 65 °C for 16 h and kept at 4 °C before preparing the cartridges. For miRNA expression, 100 ng of total RNA from each sample underwent preparation involving multiplexed annealing of specific tags onto the 3′ end of each mature miRNA, followed by ligation and enzymatic purification. miRNAs were hybridized with a code set for 12 h at 65 °C, and then the hybridized products were prepared for cartridge loading on an nCounter PrepStation. NanoString assays were performed in accordance with the manufacturer’s protocols “MAN-10023-11” and “MAN-C0009-07” (NanoString technologies).

The nCounter PanCancer Pathway Panel consists of 770 genes from 13 cancer-associated canonical pathways (https://www.nanostring.com/products/ncounter-assays-panels/oncology/ncounter-pancancer-pathways-panel/; accessed on 30 September 2021). The PanCancer Immune Profiling Panel consists of 770 genes associated with adaptive and innate immune responses from different cell types (https://www.nanostring.com/products/ncounter-assays-panels/oncology/pancancer-immune-profiling/; accessed on 30 September 2021). The NanoString nCounter miRNA Expression Assay kit measures the expression of 830 human miRNAs (https://www.nanostring.com/wp-content/uploads/2021/01/MAN-C0009-07_miRNA_Expression_Assay_User_Manual.pdf; accessed on 30 September 2021). Housekeeping genes (*n* = 40) as well as endogenous miRNAs (*n* = 5) are incorporated in the NanoString code sets and were used for analysis along with positive and negative controls. The raw data were processed using the NanoString nSolver 4.0 Software (NanoString) for quality control (QC) checks before analysis. The software provides QC flags to assess the quality of the data for imaging, binding density, linearity of positive controls, and limit of detection. The definition and implementation of this QC are summarized in nSolver documentation (MAN-C0019-08).

### 2.8. Statistical Analysis of mRNA and miRNA Expression Data

All statistical analyses were performed using the software R (R Core Team, Vienna, Austria) for statistical computing (version 4.1.0) [17]. Fold changes for all genes/miRNAs were computed as ratios of the mean among cases over the mean among controls. Given the small number of controls, a Shapiro–Wilks test [18] was used to assess the normality of the measurements within the groups. For the purpose of performing differential gene expression analysis (DGEA) of genes/miRNAs showing a *p*-value in the Shapiro–Wilks test above 5%, a *t*-test for independent samples was used to compare the means between tumors and controls; otherwise, the Mann–Whitney U test was used. Two-sided tests were used to determine the significance of the results. Given the large number of such comparisons, *p*-value adjustment for multiple testing was carried out using the Benjamin–Hochberg (B–H) method [19], which controls the false discovery rate (FDR).

The genes/miRNAs that, after the B–H correction, were found to be significant at the 5% level in discriminating tumors from controls were examined further. Principal component analysis (PCA) [20] was performed to capture the covariance structure of the data and reduce their dimension. To achieve dimensional reduction, only principal components (PCs) whose cumulated proportion of variance exceeded 80% of the total were included in the downstream analyses. For interpretation of the data covariance structure, the PCA loadings (i.e., the vectors defining the directions detected by the PCs) of the first and second PCs were examined: the genes/miRNAs showing the 5% largest (smaller) loading values in the first two PCs jointly were collected as those mostly implied in over- (under-)expression. Moreover, the PCA scores (i.e., the projections of the data on the directions detected by the PCs) of the first and second PCs were also examined to determine whether a grouping structure could be observed.

First, a scatterplot of the scores of the first two PCs with different colors for tumors and controls was inspected to look for differences between the two groups. Finally, hierarchical clustering [21] with Euclidean distance and Ward’s minimum variance linkage [22] was performed on the scores of the selected relevant PCs. The best-suited number of clusters was selected for each panel by inspection of the cluster dendrogram. Survival analysis was performed using the Kaplan–Meier method [23].

## 3. Results

### 3.1. G-Banding Analysis

All 33 ECs showed abnormal karyotypes, which was why the series was selected. Structural chromosomal rearrangements were more common than numerical aberrations. In total, 22 tumors showed a simple karyotype, meaning they displayed ≤ 5 chromosome abnormalities, whereas 11 had a complex one, meaning > 5 abnormalities or an incomplete karyotypic description. Chromosome 1 was the most commonly rearranged chromosome, being involved in 26 structural abnormalities from 23 tumors. The cytogenetic analysis revealed four distinct subgroups of ECs based on karyotypic features (Table 3): (A) tumors with an isochromosome for the long arm of chromosome 1, i(1)(q10), as the sole aberration (*n* = 5; 15%); (B) tumors showing partial duplication of the long arm of chromosome 1, dup(1)(q21q32), as the sole abnormality (*n* = 3; 11%); (C) tumors showing i(1)(q10) together with other abnormalities (*n* = 15; 45%); and (D) tumors showing no involvement/gain of/from 1q as part of the abnormal karyotype (*n* = 10; 30%).

### 3.2. Genomic Imbalances

CNAs were identified in all the 33 ECs investigated. The average number of copy aberration (ANCA) index, calculated by dividing the sum of observed copy number imbalances by the number of cases, was 26.6 [24]. The most frequent gain was of or from 1q (19 out of 33 tumors; 57%) (Figure 1A; Appendix A). This imbalance was seen in 17 out of 24 endometrioid ECs, with minimal overlapping regions found at 1q21.1, 1q23.3, and 1q25.3 in 70% of the tumors, 1q21.2q23.2, 1q24.1q25.2, and 1q31.1q32.13 in 66% (16 out of 24 tumors), and gain of 1q32.2q44 in 15 out of 24 tumors (62%) (Figure 1B). Amplifications (≥4 copies) were scored at 8p11.22 (*n* = 3). Common losses were identified on 15q11.1q11.2 and 16q21q23.1 (six tumors each or 18% of the cases), on 8p23.1, 14q32.33, 16q21, and 16q22.3 in five ECs each (15%), and on 1p36.32p36.21, 10q26.13, 16q12.2q13, and 16q24.1q24.3 in four ECs each (12%) (Figure 1A). In four out of nine nonendometrioid samples, loss from 1p (p36.32p36.21) and 16q (q23.3q22.1) were identified (44%) (Figure 1C).

### 3.3. Pathogenic Variants

Fifty genes were investigated for presence of pathogenic variants using the Ion Torrent platform. All ECs but one (case 1) gave informative results; however, tumors 19 and 24 did not show any pathogenic variants (Figure 2; Appendix A). Activating mutations in *PTEN* were seen in 17 ECs (53% of the total number of informative cases), followed by mutations in *PDGFRA* (*n* = 12, 37%), *PIK3CA* (*n* = 11, 34%), *KIT* (*n* = 10, 31%), *CTNNB1* (*n* = 6, 19%), *FGFR2* and *KRAS* (both *n* = 5, 16%), *TP53*, and *FBXW7* (both *n* = 3, 9%), APC, *ATM*, and *MET* (*n* = 2, 7%), and *IDH1* (*n* = 1, 3%).

None of the samples showed mutations in the *POLE* gene in any of the five investigated hotspots/loci.

### 3.4. Microsatellite Instability (MSI)

MSI-H was identified in 15 ECs (45% of the samples) and MSI-L in 3 (9%), whereas the remaining 15 samples were MSS (45.5%). The majority of MSI-H cases (*n* = 12) showed more than five unstable markers (Table 4).

### 3.5. Gene Expression Profile—PCA, DGEA, and Hierarchical Clustering

DGEA with multiple testing corrections via the B–H method, PCA, and subsequent hierarchical clustering provided an overview of the most relevant gene expression profiles for each panel, i.e., PanCancer Pathway, Immune Profiling, and miRNA (Figure 3). In all panels used, the controls clustered separately from the tumors. A total of 478 genes were found differentially expressed after the B–H correction for comparing ECs and normal tissues when the PanCancer Pathway set was used. After running a PCA on the B–H-adjusted (*p* < 0.05 and *p*-adj < 0.05) significantly upregulated genes in ECs (compared with controls), 10 genes were found to be in the top-5% upregulated, whereas 13 genes were found in the top-5% downregulated in the PC loadings. The genes found significantly upregulated belonged to the apoptosis pathway (*PIKMYT;* fivefold change), the DNA repair pathway (*BRCA1*, *POLR2H*, *POLR2D*, *FANCG*; threefold change), the MAPK/PI3K pathway, and the transcriptional regulation pathway (*NF1*, *DAXX*, and *MEN1*; two-\fold change). The *FUB1* and *MSH2* genes, which were found to be upregulated, are classified as cancer drivers in the NanoString platform and in the latest studies [25,26]. The downregulated genes are known to be involved/participate in the MAPK/PI3K pathway (*HGF*, *PLA2G4A*, *PDGFRA*, *FGF2*, *GADD45G*, *ITGA8*, *FGF7*, and *RASGRP2*), in the transcriptional regulation pathway (*ZBTB16*, *NGFR*, *RUNX1T1*), in the RAS pathway (*SHC2*, *HGF*, *FGF7*, *RASGRP2*), and in apoptosis (*GADD45G*). The most downregulated genes were *HGF* and *GADD45G* (Appendix A).

In total, 199 genes were differentially expressed comparing EC samples and controls using the Immune Profiling Panel; of these, 5 were significantly upregulated and 10 significantly downregulated (Appendix A). *GPI* was the most commonly upregulated gene (threefold change), followed by *CASP3*, *TYK2*, *EWSR1*, and *TBK1* (fold change between 1.6 and 2.1). Among the downregulated genes, *CCL19*, *NCAM1*, *AKT3*, *ABCB1*, *CFD*, *MASP1*, *C7*, *CCL21*, *CCL14*, and *CXCL12* were the most commonly involved (fold change between 0.02 and 0.48).

A total of 75 miRNAs were found differentially expressed between ECs and controls. Four miRNAs were significantly upregulated (miR-107, miR-148-3p, miR-93-5p, and miR-32-5p) with fold changes between 2.45 and 15.12, whereas three were significantly downregulated (miR1247-5p, miR-875-3p, and miR-455-3p) with a fold change between 0.18 and 0.51 (Appendix A).

### 3.6. Statistical Correlation

The nonparametric Mann–Whitney U test for statistical analysis was used to compare the four cytogenetic subgroups with FIGO stage, post-operative treatment, response, recurrence, death, and OS. No significant correlation was observed.

## 4. Discussion

ECs are heterogeneous both among tumors and within each tumor. Furthermore, it is well-known that even benign endometrial tumors such as atypical endometrial hyperplasia can share molecular alterations with tumors that are already malignant [27,28,29], features that have an important impact on diagnostic/therapeutic decisions [7]. Besides this false-positive type of similarity, precise diagnoses are made difficult by the fact that many ECs lack specific molecular profiles [9,30].

Incorporating molecular parameters identified by the TCGA [10] into clinical practice has been useful when it comes to assessing prognostic parameters in grade 3 endometrioid ECs; however, many challenges remain. Applying the new classification based on TCGA findings to our series of ECs, we could identify only two molecular subgroups of the four found in the original study [10]: one characterized by high CNAs (*n* = 28; >five genomic imbalances) and the other by MSI-H (*n* = 15). Interestingly, all but two tumors (cases 11 and 13) of the MSI-H subset also showed CNA-H, i.e., the two groups overlapped. Five tumors could not be classified according to the TGCA or the WHO criteria. Similar difficulties in EC classification are well-known and characterized also the series examined by Kandotch et al. [10]. Additionally, the latest studies suggested the importance of including TCGA molecular subgroups to assess the prognosis of ECs in association with clinical–pathological factors [31,32,33].

Cytogenetic data on ECs show that hyperdiploid karyotypes with numerical aberrations and/or simple structural rearrangements characterize two-thirds of tumors, whereas the remaining one-third have complex karyotypes with several numerical and structural chromosome abnormalities [12,34]. Our series is in line with previously published data inasmuch as chromosome 1 was found to be the most commonly rearranged chromosome: 45% of the tumors showed i(1)(q10), often together with other abnormalities. Only a small group of samples (*n* = 8) showed the involvement of chromosome 1 as the sole karyotypic abnormality. This finding was confirmed by aCGH data showing recurrent gains of or from 1q. According to the Mitelman database, 1q rearrangements are often seen as the sole change in carcinoma of the uterine corpus (34 cases out of 115 cases scientifically reported) [12]. Milatovic et al. [35] and Micci et al. [24] suggested that the 1q aberration was primary in a subset of these tumors that subsequently accumulated additional chromosomal aberrations during clonal evolution. In view of this hypothesis, we investigated whether the four cytogenetic subgroups recognized in this study correlated with the disease parameters tumor histology, FIGO stage, post-operative treatment, response, recurrence, death, and overall survival (OS, calculated by the date of histological diagnosis until the death or last update). No significant chromosomal–clinical correlation could be identified.

Intra-tumor heterogeneity, seen as the presence of multiple clones, was not identified in this series. This cytogenetic uniformity strongly supports the notion that ECs are of monoclonal origin. The fact that material for all analyses—be it at the chromosomal, DNA, or RNA level—were from the same small sample is also unique to this study, as is indeed the very fact that all these three fundamental organizational levels were genetically examined. At the same time, some intrinsic limitations are equally obvious, not least the limited size of the cohort. It would be of value in the future to enlarge the number of examined tumors, both endometrioid and nonendometrioid ECs. In particular, having more tumors of the serous, clear cell, and mucinous subtypes available for analysis would probably be informative with regard to assessing the heterogeneity of these subgroups. Due to the fact that only small amounts of tumor material were available, it was not possible to run tests in duplicate or triplicate.

At the molecular level, a high frequency of pathogenic variants was observed in the genes *PTEN* (53%), *PDGFRA* (37%), *PIK3CA* (34%), *KIT* (31%), and *CTNNB1* (19%). The mutation rates for our samples only slightly differed from those previously found [10,36]. By way of example, *PTEN* was reported by the COSMIC database to show variants in 43% of cases, *PIK3CA* in 28%, and *CTNNB1* in 19% [36]. The differences between our series and the reported frequencies could be explained by the use of “search filters”. The COSMIC database applied “endometrium” and “all carcinoma” as filters, meaning that entities such as carcinosarcomas become part of the final frequencies, something that could have led to slightly different frequencies from what would be the case if only ECs were examined.

Perhaps interestingly, we did not find any pathogenic variants of the *POLE* gene, although this has been reported at frequencies of 7–12% in ECs [11,37,38]; this should correspond to 1–3 cases with pathogenic variants in our series. Most likely, the fact that none of our samples had them is due to chance.

Microsatellite status is a strong prognostic marker in patients with colorectal cancer [39,40]. Lately, this biological feature has been used to define also a subgroup of endometrioid ECs [10,41], and its prognostic value has been evaluated extensively [42,43]. Some studies have shown better survival, others poor survival, whereas others again have found no association at all [43,44,45]. In our series, MSI-H was found in 48% of ECs and present in patients with OS of 48.64 months (approximately four years; data updated to the year 2021; *p*-value = 0.85).

The transcriptomic/expression studies were approached using unsupervised cluster and PCA analysis. The fact that all controls clustered separately from the tumor samples speak of the reliability of the dataset. This notwithstanding, substantial heterogeneity was observed among the tumors, and we found no meaningful sub-clustering based on clinical–pathological–molecular parameters. The most significant upregulated mRNA in the present cohort was the protein kinase membrane associated tyrosine/threonine 1 (*PKMYT1*), whereas hepatocyte growth factor (*HGF*) followed by growth arrest and DNA damage-inducible gamma (*GADD45G*) were the most downregulated. These genes also showed the largest fold-change value. *PKMYT1* was the most upregulated gene in our study. It maps to chromosome sub-band 16p13.3 and belongs to the Ser/Thr family of protein kinases (WEE1) [46]. The gene encodes a cyclin-regulating kinase that inhibits phosphorylation of cyclin-dependent kinase 1 (Cdk1), a master regulator of the cell cycle. *PIKMT1* might be expected to act as a tumor suppressor by preventing Cdk1 activation. The abnormal expression of *PKMYT1* has been associated with soft tissue sarcoma, prostate adenocarcinoma, and squamous cell carcinoma of the cervix, among other neoplasms [36,47], although its role in tumorigenesis still has not been thoroughly investigated. *HGF* maps to chromosomal band 7q21.1 and regulates cell growth, motility, and morphogenesis in numerous cell and tissue types [48]. Deregulation of growth factors is a well-known characteristic of many neoplastic/cancerous cells [36,49,50,51,52]; it is likely to be a feature also of uterine carcinomas. Activation of the HGF pathway can lead to increased cell survival, proliferation, and the metastatic spreading of cancer cells [53]. It has been shown that aberrant activation of the HGF pathway in tumor tissues can predict drug response [54]. Different preclinical studies have shown that the inhibition of HGF activity is required to overcome therapeutic resistance in amplified cancer cells, thus increasing cancer patients’ overall survival [55].

*GADD45G*, which maps to chromosome sub-band 9q22.2, is a member of a group of genes whose increased transcription in response to stress shock can inhibit cell growth and induce apoptosis in many solid cancers as well as hematopoietic malignancies [56]. The gene is an essential player in the oncogenesis of thyroid carcinoma, pancreatic ductal carcinoma, and colon carcinoma [36,57,58] but is now, for the first time, shown to be deregulated also in ECs. Its role in the pathogenesis of these cancers should be further investigated, preferably in larger cohorts.

The upregulated genes were involved in apoptosis, DNA repair, and transcriptional regulation, whereas the downregulated genes were involved in MAPK/PI3K, Ras, transcriptional regulation, and apoptosis pathways.

Expression analysis based on immune profiling demonstrated a significant change for glucose-6-phosphate isomerase (*GPI*) (fold change 3.2). *GPI* maps to 19q13.11 and is a member of the glucose phosphate isomerase family that plays a role in glycolysis and gluconeogenesis. It may function as a cytokine and angiogenic factor. The aberrant expression of *GPI* has been involved in the development and progression of several cancers, such as adrenal cortical carcinoma, lung adenocarcinoma, and malignant melanoma, possibly by influencing immune cell infiltration [36,59,60].

miRNAs are small noncoding RNAs that regulate gene expression in a variety of manners, including translational repression, mRNA cleavage, and deadenylation. They can play a role in tumor development and/or progression by regulating the expression pattern of oncogenes and suppressor genes [61,62]. The most upregulated miRNA found in our study was miR-32-5p (fold change 15.1), which maps to Xq26.2, is a member of the miR-32 family, and is involved in the development of numerous cancers. miR-32-5p plays a key role in the tumorigenesis of various types of cancer by regulating multiple target genes [63]. Liu et al. [64] showed that miR-32-5p is upregulated in cervical carcinoma tissues, where it inhibits cell proliferation, migration, and invasion. Moreover, numerous miRNAs, including miR-32-5p, interact functionally with *PTEN* inhibiting its expression. *PTEN* was the most mutated gene in our series. Out of the 17 EC samples with pathogenic *PTEN* variants, 13 showed an aberrant expression of miR-32-5p. Fu et al. [65] showed that miR-32-5p could activate the PI3K/Akt pathway and induce multidrug resistance by modulating angiogenesis and the epithelial–mesenchymal transition (EMT) process.

The two samples classified as “adenocarcinoma NOS” shared the same heterogeneous profile as all EC tumors regarding pathogenic variants, as well as genes/miRNAs expression. No subtype-specific clusters were identified. This phenomenon has been seen before [8,10] with most serous and mixed histology tumors (i.e., nonendometrioid carcinomas) clustering together with some of the endometrioid tumors. In our series, it was not possible to determine any specific genomic/transcriptomic profile that set endometrioid apart from nonendometrioid carcinomas.

## 5. Conclusions

Also, in the present study, a high degree of heterogeneity existed among ECs even by unsupervised analysis of expression profiles. This notwithstanding, tumor samples were as a group well-separated from controls, albeit with no clear-cut separation between endometrioid and nonendometrioid ECs. Whether this means that the current phenotypic classification is of little relevance or if one still has not detected which genomic parameters to enter into correlation analyses is still unknown. Regardless, a better classification based on relevant biological tumor features is clearly still needed for carcinomas of the uterine corpus, preferably one that takes into account the essential genomic events of carcinogenesis.

## Figures and Tables

**Figure 1 cancers-14-03536-f001:**
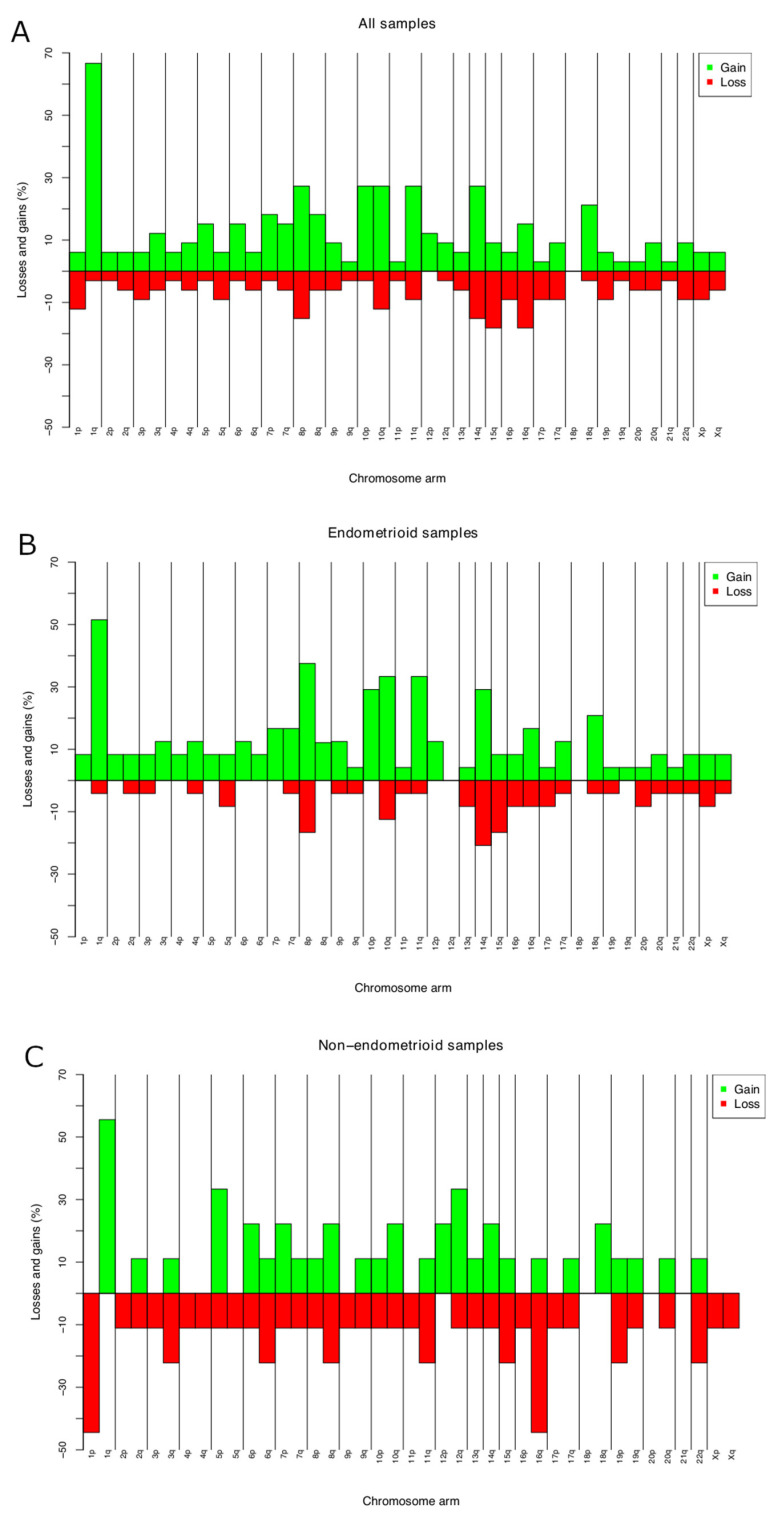
Chromosome imbalances identified by aCGH in 33 primary ECs (**A**); chromosome. imbalances in 24 endometrioid ECs (**B**); and 9 nonendometrioid ECs (**C**). Frequencies reported are the highest for the entire arms.

**Figure 2 cancers-14-03536-f002:**
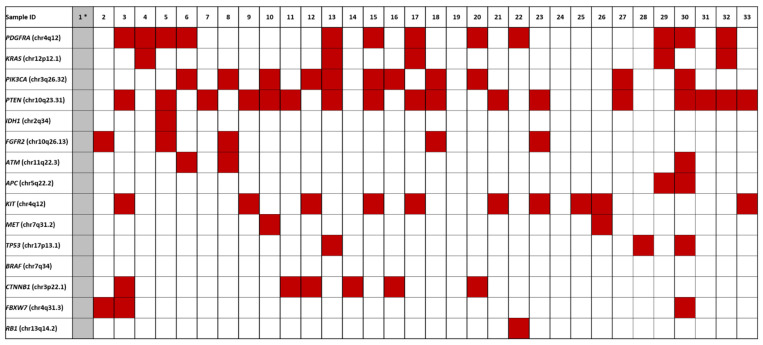
Overview of the pathogenic variants identified in ECs. Case 1 *, colored in grey, was not informative.

**Figure 3 cancers-14-03536-f003:**
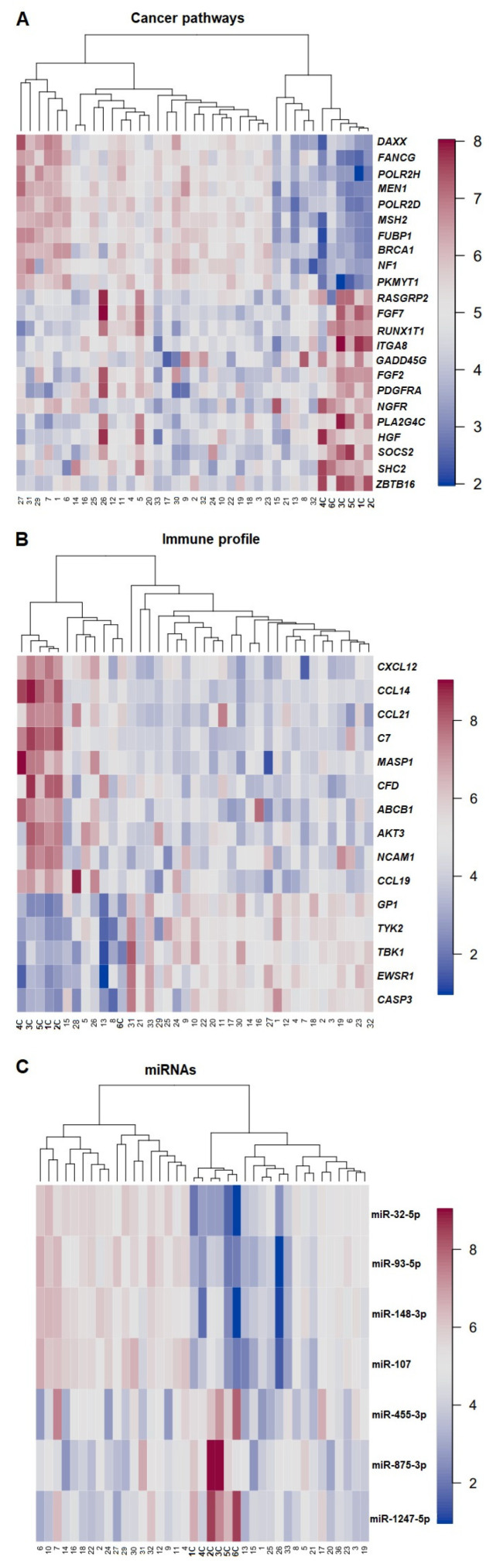
Heatmap of expression profiles raw data for cancer pathways (**A**), immune profile (**B**), and miRNAs (**C**). Samples are arranged in columns and the genes, and miRNAs expression levels are arranged in rows. The blue shades indicate reduced expression, and the red shades indicate increased expression. Numbers in bold highlight control samples.

**Table 1 cancers-14-03536-t001:** Clinicopathological characteristics of the material (*n* = 33).

Characteristics	Distribution
Post-operative histology	Endometrioid G1 (*n* = 8) Endometrioid G2 (*n* = 11) Endometrioid G3 (*n* = 5)
	Clear Cell (*n* = 1) Serous (*n* = 3) Mixed (*n* = 3) Adenocarcinoma NOS (*n* = 2)
FIGO Stage 2009	IA (*n* = 13) IB (*n* = 9) II (*n* = 2) IIIA (*n* = 2) IIIB (*n* = 1) IIIC1 (*n* = 2) IVB (*n* = 3)
Post-operative treatment	none (*n* = 20) Platinum based chemotherapy (*n* = 12) Chemotherapy + radiation therapy (*n* = 1)
Recurrence	yes (*n* = 10) no (*n* = 23)
Site of recurrence	vagina (*n* = 3) distant (*n* = 2) multiple sites (*n* = 4) isolated nodal pelvic (*n* = 1)
Age at diagnosis	average years 70 (range 54 to 94)
Deaths	*n* = 9
OS	average months 44.5

FIGO, International Federation of Gynecology and Obstetrics; G, grade; NOS, not otherwise specified; OS, overall survival.

**Table 2 cancers-14-03536-t002:** Primers used for PCR reactions.

Name	Sequence	Gene	Position on GRch37/hg19 Assembly
**M13**-POLEint9FW	5′-**TGTAAAACGACGGCCAGT**AGTCTTAGGGTCCTTCTCCCA-3′	*POLE*	chr12 − 132676775 132676795
**M13**-POLEint10R	5′-**CAGGAAACAGCTATGACC**TGTGTGGATTCCCACTCGAAA-3′	*POLE*	chr12 + 132676361 132676381
**M13**-POLEint12FW3	5′-**TGTAAAACGACGGCCAGT**GGGGTTCCCGGGCTGCATGTTA-3′	*POLE*	chr12 − 132673744 132673765
**M13**-POLEint13REV3	5′-**CAGGAAACAGCTATGACC**CTCCGTGGCCATCTGGATGCGT-3′	*POLE*	chr12 + 132673513 132673534
**M13**-POLEint13F	5′-**TGTAAAACGACGGCCAGT**ACACACGTGTTTTGTCCTGTG-3′	*POLE*	chr12 − 132673397 132673417
**M13**-POLEint14R	5′-**CAGGAAACAGCTATGACC**CAGGGCCAGAGAATTCCCAA-3′	*POLE*	chr12 + 132672996 132673015

**Table 3 cancers-14-03536-t003:** Overview of the karyotypes for ECs.

Sample	Karyotype	Subgroup *
1	46,XX,dup(1)(q21q32)[15]/46,XX[1]	B
2	48,XX,+i(1)(q10),+?10[10]/46,XX[8]	C
3	47,XX,+i(1)(q10),t(8;16)(p21;p13)[cp11]	C
4	48~49,XX,+i(1)(q10),+2,inc[cp4]/46,XX[17]	C
5	49~50,XX,+X,+i(1)(q(10),+10[cp4]/46,XX[2]	C
6	47,XX,+i(1)(q10)[3]	A
7	53-55,XX,+del(1)(p31),+i(1)(q10),+3,+6,+7,+8,+8,+9,−13,−15,+2mar[cp10]	C
8	47,XX,+der(1;10)(q10;q10)[9]/46,XX[1]	C
9	41~42,der(19)t(1;19)(q21;q13),inc[cp2]/46,XX[6]	C
10	46,XX,add(3)(q21)[cp2]/46,XX[23]	D
11	47,XX,+i(1)(q10)[7]/46,XX[3]	A
12	46,XX,del(16)(q12)[11]/46,XX[4]	D
13	46,XX,dup(1)(q23q32)[cp8]/46,XX[2]	B
14	50~100,+i(1)(q10),+7,inc[cp9]/46,XX[1]	C
15	48,XX,+9,+10,der(15)t(1;15)(q11;p13)[8]/46,XX[2]	C
16	46~47,XX,+i(1)(q10)[cp6]/46,XX[4]	A
17	45,XX,t(12;14)(q15;q24),-22[6]/43~45,idem,+r,inc[cp2]/46,XX[2]	D
18	42~46,XX,+7,-21[cp7]/91~93,idemx2[cp3]/46,XX[4]	D
19	47,XX,+X[11]	D
20	47,XX,+1,der(1;13)(q10;q10),+8[10]	C
21	47,XX,+add(1)(p11)[5]/46,XX[5]	A
22	47,XX,+10,del(16)(q13)[3]/46,XX[8]	D
23	46,XX,r(13)[10]	D
24	41~70,i(1)(q10),+mar,inc[cp7]/46,XX[1]	C
25	46,XX,t(2;12)(q37;q?13)[3]/46,XX[9]	D
26	46,XX,?t(4;6)(q?28;p22),add(11)(q22)[6]/46,XX[5]	D
27	52,XX,i(1)(q10),+3,+8,+8,+10,+10[10]	C
28	38~75,inv(1)(p13p36),add(2)(q33),add(7)(q32),inc[cp7]/46,XX[5]	D
29	36~74,dup(1)(q21q23),der(?)t(1;?)(p13;?),der(?)t(1;?)(q12;?),add(11)(q23),inc[cp7/46,XX[4]	B
30	44~46,XX,der(1)t(1;1)(p36;q21~q23)[cp7]/46,XX[3]	A
31	44~46,add(1)(p13),+i(1)(q10),del(3)(q13),inc[cp13]	C
32	48~49,XX,+del(1)(p22),+12,inc[cp6]	C
33	45~47,XX,der(3)t(1;3)(q11;p25)[cp3]/46,XX[7]	C

* (A) tumors with an isochromosome for the long arm of chromosome 1, i(1)(q10), as the sole aberration (*n* = 5); (B) tumors showing partial duplication of the long arm of chromosome 1, dup(1)(q21q32), as the sole abnormality (*n* = 3); (C) tumors showing i(1)(q10) together with other abnormalities (*n* = 15); (D) tumors showing no involvement/gain of/from 1q as part of the abnormal karyotype (*n* = 10).

**Table 4 cancers-14-03536-t004:** Overview of the MSI findings.

Sample	Overall Call	Markers Unstable	Markers Stable
1	MSS	0	13
2	MSI-Low	1	12
3	MSI-High	5	8
4	MSI-High	10	3
5	MSI-High	6	7
6	MSI-High	5	8
7	MSI-Low	2	11
8	MSI-High	5	8
9	MSI-Low	1	12
10	MSS	0	13
11	MSI-High	12	1
12	MSI-High	6	7
13	MSI-High	4	9
14	MSS	0	13
15	MSS	0	13
16	MSS	0	13
17	MSI-High	7	6
18	MSI-High	10	3
19	MSI-High	7	6
20	MSI-High	4	9
21	MSI-High	7	6
22	MSI-High	6	7
23	MSS	0	13
24	MSS	0	13
25	MSS	0	13
26	MSS	0	13
27	MSS	0	13
28	MSS	0	13
29	MSS	0	13
30	MSI-High	7	6
31	MSS	0	13
32	MSS	0	13
33	MSS	0	13

## Data Availability

The data presented in this study are available on request from the corresponding author.

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
