# Peer review of "Endometrial Carcinoma: Molecular Cytogenetics and Transcriptomic Profile"

_cancers, 2022, doi:10.3390/cancers14143536_

Round 1
Reviewer 1 Report
In this study 33 samples from endometrial cancers and 6 normal controls have been analysed regareing chromosomal aberrations, genomic imbalances, pathogenic variants, microsatellite instability and expression profiles at gene and miRNA level.
The analyses are extensive but the sample rather small. The method section needs some clarifications, and do not discuss findings that has not been displayed in the result section.
Following are more specific comments:
Introduction page 2 line ...? (manuscript contains no lines) «endometrial carcinomas (EC) which ac- count for about 90% of malignancies in post-menopausal women» this is slightly inaccurat, malignancies being a broad term including more than uterine cancers and breast cancer is a more common malignancy in females than endometrial cancer.
Materials:
Which type of study is this? Retrospective cohort? The selection of these patient samples, what were they based upon/why were these chosen? Where they a specific patient series collected at this department? A specific study started in 2011 as the Research Ethics number is including the Year 2011? I this cohort representative of the departments population? Adding age as a variable in table 1 would be appreciated.
The «controls», which tissue/cells/? were these? Normal endometial cell lines? This MUST be explained.
Post-operation treatment, this is usually named either post-operative or adjuvant treatment.
Describe all abbreviations , both first time in text and for all/each table and figures. E.g FIGO, G1, NOS, OS
In methods: SNP and SNA should also be spelled out
For all materials used, name company, city and country for main office should be stated.
Were these analyses performed as one per tumor or were analyses performed in multiple (duplicate/triplicate?) incorporting possibility of tumor heterogeneity?
Statistics: I appreciate that correction for multiple testing has been considered.
Were tests one- or two-sided? (not stated)
Ethic statement is missing: did the patients consent to this study or was this not deemed nessesary? Please specify.
Results:
In table 3 the different subgroups (A, B, C, D) needs to be explained either in legend or as footnote. Also the number of patients included with these finding should be added.
EACH table/figure should be possible for the reader to understand without reading the full paper.
Page 10 «The FUB1 and MSH2 genes, which were found to be upregulated, are known to be cancer drivers (NanoString technologies)». Is this supposed to be a reference? Please add approprietly.
Figure 3, asuming the numbers below signifies individual specimen/patients, this should be explained. Nubers added with C are perhaps controls? This should also be explained.
Discussion:
«In view of this hypothesis, we investigated whether the four cytogenetic subgroups recognized in this study correlated with the disease parameters tumor histology, FIGO stage, post-operative treatment, response, recurrence, death, and overall survival (OS, calculated by the date of histological diagnosis until the death or last update). No significant chromosomal-clinical correlation could be identified.»
This statement is not supported by data shown in the result section. Table 1 (in the method section) show the basic clinicopathological data for all patients but no recurrence/survival data have been displayed and I do not find any reference to survial statistics in the method section.
The discussion does not include any comments regarding strength or limitations of the study.
Author Response
Oslo, 12th July 2022
Dear Editor,
We were pleased to receive the comments of the reviewers on our manuscript entitled “Endometrial Carcinoma: molecular cytogenetics and transcriptomic profile” (ID nr Cancers-1789360).
Enclosed, please find a revised version in which we have taken all suggestions/criticism into account. We hope the revised manuscript is acceptable for publication in Cancers.
Below is a point-to-point answer to the reviewers’ comments. Our answers are in italics.
Reviewer #1:
Introduction page 2 «endometrial carcinomas (EC) which account for about 90% of malignancies in post-menopausal women» this is slightly inaccurat, malignancies being a broad term including more than uterine cancers and breast cancer is a more common malignancy in females than endometrial cancer.
- We agree with the reviewer that the sentence was badly formulated and, in the end, inaccurate. We have now reformulated it in the following way “The most frequent uterine cancers are endometrial carcinomas (EC) which account for 90% of uterine malignancies in post-menopausal women [3].”
Materials: Which type of study is this? Retrospective cohort? The selection of these patient samples, what were they based upon/why were these chosen? Where they a specific patient series collected at this department? A specific study started in 2011 as the Research Ethics number is including the Year 2011? I this cohort representative of the departments population?
- The study is based on a retrospective cohort for which we had karyotypic data. We have specified this in the results section under paragraph 3.1 G-banding analysis, adding the following sentence: “All 33 EC showed abnormal karyotypes, that was why the series was selected.”
Adding age as a variable in table 1 would be appreciated.
- We have added the information about the age in Table 1, both as average and range.
The «controls», which tissue/cells/? were these? Normal endometial cell lines? This MUST be explained.
- We are grateful for the comment and have now specified the type of controls. The following sentence was added to the Material and Methods section, paragraph 2.1 Tumor material: “Six commercially available normal controls were included in the series. Four of them were total RNA from the normal uterus of single donors (AMS Biotechnology, Abingdon, United Kingdom). The fifth was total RNA consisting of a pool of uterine RNA from five donors (AMS Biotechnology). The sixth was from normal endometrium from a single donor (OriGene, Rockville, Maryland, USA).”
Post-operation treatment, this is usually named either post-operative or adjuvant treatment.
- We are sorry for the error, which is now corrected in Table 1 as Post-operative treatment.
Describe all abbreviations, both first time in text and for all/each table and figures. E.g FIGO, G1, NOS, OS
- We have now added a footnote in Table 1 specifying all abbreviations.
In methods: SNP and SNA should also be spelled out.
- We have used the long terminology for the mentioned abbreviations, and these are now described in paragraph 2.4. Genomic imbalances.
For all materials used, name company, city and country for main office should be stated.
- As suggested, we have double-checked all materials used, which are now described with the company name, city, and country throughout the manuscript.
Were these analyses performed as one per tumor or were analyses performed in multiple (duplicate/triplicate?) incorporting possibility of tumor heterogeneity?
- We are grateful for the comment. We have now specified the approach in a better way, adding the following paragraph under the Discussion section (page 13) “Intra-tumor heterogeneity, seen as presence of multiple clones, was not identified in this series. This cytogenetic uniformity strongly supports the notion that EC are of monoclonal origin. The fact that material for all analyses - be it at the chromosomal, DNA or RNA level - were from the same small sample, is also unique to this study, as is indeed the very fact that all these three fundamental organizational levels were genetically examined. At the same time, some intrinsic limitations are equally obvious, not least the limited size of the cohort. It would be of value in the future to enlarge the number of tumors examined, both endometrioid and non-endometrioid EC. In particular, to have more tumors of the serous, clear cell, and mucinous subtypes available for analysis would probably be informative with regard to assessing the heterogeneity of these subgroups. Due to the fact that only small amounts of tumor material were available, it was not possible to run tests in duplicate or triplicate.”
Statistics: I appreciate that correction for multiple testing has been considered. Were tests one- or two-sided? (not stated).
- The following sentence was added under paragraph 2.8 to clarify the test used: “Two-sided tests were used to determine the significance of the results.”
Ethic statement is missing: did the patients consent to this study or was this not deemed nessesary? Please specify.
- The following sentence was included at the end of the manuscript (page 16) at the time of submission “Informed consent statement: Informed consent was obtained from all subjects involved in the study.” We believe that this is according to the Guidelines of the journal; we have therefore not added further specification on the subject to avoid repetitions.
Results: In table 3 the different subgroups (A, B, C, D) needs to be explained either in legend or as footnote. Also the number of patients included with these finding should be added. EACH table/figure should be possible for the reader to understand without reading the full paper.
- We have now specified the different subgroups and the number of patients in each group as a footnote in Table 3.
- In line of this comment, we have also added the following sentence to the legend for Figure 2 “Case 1*, coloured in grey, was not informative.”
Page 10 «The FUB1 and MSH2 genes, which were found to be upregulated, are known to be cancer drivers (NanoString technologies)». Is this supposed to be a reference? Please add approprietly.
- The mentioned sentence was reformulated to make it clearer, and two references were also added. Now it reads as “The FUB1 and MSH2 genes, which were found to be upregulated, are classified as cancer drivers in NanoString platform and in the latest studies [25,26].”
Figure 3, asuming the numbers below signifies individual specimen/patients, this should be explained. Nubers added with C are perhaps controls? This should also be explained.
- The Figure legend for Figure 3 is now modified as follows “Heatmap of expression profiles raw data for cancer pathways (A), immune profile (B), and miRNAs (C). Samples are arranged in columns and the genes and miRNAs expression levels in rows. The blue shades indicate reduced expression and the red shades indicate increased expression. Numbers in bold highlight control samples. Since we introduced bold numbers for controls, a new figure is enclosed in the re-submission (Figure 3. revised).
Discussion: «In view of this hypothesis, we investigated whether the four cytogenetic subgroups recognized in this study correlated with the disease parameters tumor histology, FIGO stage, post-operative treatment, response, recurrence, death, and overall survival (OS, calculated by the date of histological diagnosis until the death or last update). No significant chromosomal-clinical correlation could be identified.»
This statement is not supported by data shown in the result section. Table 1 (in the method section) show the basic clinicopathological data for all patients but no recurrence/survival data have been displayed and I do not find any reference to survial statistics in the method section.
- We have now added a new paragraph in the Results section to make the findings clearer.
3.6. Statistical correlation
The non-parametric Mann-Whitney U Test for statistical analysis was used to compare the four cytogenetic subgroups with FIGO stage, post-operative treatment, response, recurrence, death, and OS. No significant correlation was seen.
- The following sentence is present in Material and Methods at the end of paragraph 2.8. Statistical analysis of mRNA and miRNA expression data section “Survival analysis was performed using the Kaplan–Meier method [23].”
The discussion does not include any comments regarding strength or limitations of the study.
- We have now a sentence on this subject in the Discussion. See answer to a previous point where the following sentence was added “Intra-tumor heterogeneity, seen as presence of multiple clones, was not identified in this series. This cytogenetic uniformity strongly supports the notion that EC are of monoclonal origin. The fact that material for all analyses - be it at the chromosomal, DNA or RNA level - were from the same small sample, is also unique to this study, as is indeed the very fact that all these three fundamental organizational levels were genetically examined. At the same time, some intrinsic limitations are equally obvious, not least the limited size of the cohort. It would be of value in the future to enlarge the number of tumors examined, both endometrioid and non-endometrioid EC. In particular, to have more tumors of the serous, clear cell, and mucinous subtypes available for analysis would probably be informative with regard to assessing the heterogeneity of these subgroups.”
Trusting you will find these responses satisfactory, I remain
Yours sincerely,
Marta Brunetti
Reviewer 2 Report
This article reports some interesting data about the karyotypic features, genomic imbalances, pathogenic variants, microsatellite instability, and expression profiles at the gene and miRNA level for 33 cases of endometrial carcinoma (EC).
To date, TCGA molecular factors have been demonstrated to be accurate in the prediction of the prognosis of EC have been associated to classic pathologic risk factors (such as grade, stage, myometrial invasion or LVSI), as reported in the ESGO/ESTRO/ESP guidelines for the management of EC. This study proposes some new information about molecular characterization pf EC, worth to be further analyzed, with the potential to be also a therapeutic target.
I have the following comments to the Authors:
· In chapter 3.4, please correct MSS-H. It should be MSI-H instead, as shown in table 4.
· Materials and methods: In order to make the study reproducible, Authors should describe how were selected the patients included in the study and how was selection bias excluded during this phase.
· Discussion: In this study, Authors reported data about the karyotypic features, genomic imbalances, pathogenic variants, microsatellite instability, and expression profiles at the gene and miRNA level for 33 cases EC. The ESTRO/ESGO/ESP guidelines for the management of EC proposed a novel risk stratification model including molecular TCGA molecular groups to assess the prognosis of EC in association with classic, well-known, clinicopathologic prognostic factors of EC (such as myometrial invasion, histotype or lymph vascular space invasion). Some recent studies analyzed the relationship between TCGA groups and classic prognostic factors (grade, myometrial invasion, LVSI) and I believe the discussion should include those recent findings in order to add also a clinical point of view to the article (e.g. PMID: 34088515; PMID: 35078650).
Author Response
Oslo, 12th July 2022
Dear Editor,
We were pleased to receive the comments of the reviewers on our manuscript entitled “Endometrial Carcinoma: molecular cytogenetics and transcriptomic profile” (ID nr Cancers-1789360).
Enclosed, please find a revised version in which we have taken all suggestions/criticism into account. We hope the revised manuscript is acceptable for publication in Cancers.
Below is a point-to-point answer to the reviewers’ comments. Our answers are in italics.
Reviewer #2:
In chapter 3.4, please correct MSS-H. It should be MSI-H instead, as shown in table 4.
- We are grateful for the comments. The correct form, MSI-H, is now included.
Materials and methods: In order to make the study reproducible, Authors should describe how were selected the patients included in the study and how was selection bias excluded during this phase.
- We have specified the inclusion criteria in the result section under paragraph 3.1 G-banding analysis, adding the following sentence: “All 33 EC showed abnormal karyotypes, that was why the series was selected.”
Discussion: In this study, Authors reported data about the karyotypic features, genomic imbalances, pathogenic variants, microsatellite instability, and expression profiles at the gene and miRNA level for 33 cases EC. The ESTRO/ESGO/ESP guidelines for the management of EC proposed a novel risk stratification model including molecular TCGA molecular groups to assess the prognosis of EC in association with classic, well-known, clinicopathologic prognostic factors of EC (such as myometrial invasion, histotype or lymph vascular space invasion). Some recent studies analyzed the relationship between TCGA groups and classic prognostic factors (grade, myometrial invasion, LVSI) and I believe the discussion should include those recent findings in order to add also a clinical point of view to the article (e.g. PMID: 34088515; PMID: 35078650).
- We agree with the reviewer that different parameters should be evaluated for the risk stratification of each patient (molecular groups, clinicopathological information, as well as prognostic factors). We have added the following sentence in the Discussion section:“Also the latest studies suggested the importance of including TCGA molecular subgroups to assess the prognosis of EC in association to clinical-pathological factors [31-33].”
Trusting you will find these responses satisfactory, I remain
Yours sincerely,
Marta Brunetti